# Analysis of Composite Coating of Deep Drawing Tool

**Jan Novotný [1],\*, Iryna Hren [1],\*, Štefan Michna [1] and Stanislaw Legutko [2]**

1   Faculty of Mechanical Engineering, J. E. Purkyne University in Usti nad Labem,
    400 01 Usti nad Labem, Czech Republic; stefan.michna@ujep.cz
2   Faculty of Mechanical Engineering, Poznan University of Technology, 60-965 Poznan, Poland;
    stanislaw.legutko@put.poznan.pl
\*   Correspondence: jan.novotny@ujep.cz (J.N.); iryna.hren@ujep.cz (I.H.)

**Abstract:** Modern coating methods have become an important part of industrial practice. For some materials and operations, the use of abrasion-resistant and hard coatings is an absolute necessity; for others, they are the key to greater efficiency and productivity. The aim of this work was to apply and subsequently analyze a new type of thin coating micro-layers TiAlN and TiAlCN, applied using HIPIMS coating technology from a physical point of view. In particular, chemical composition (EDS) and microstructure analyses were carried out in the area of applied coatings. Prepared cross-sectional metallographic samples were evaluated using electron microscopy. A detailed microstructural characterization of the individual elements was carried out on the lamellae of the investigated sample using transmission electron microscopy. It was found that this new multilayer micro-coating based on TiAlN + TiAlCN at a thickness of 5.8 μm increases the repeatability of production strokes by 200%. This finding was confirmed by testing the production of cartridges in the real operation of a large manufacturing company.

**Keywords:** coatings; TiAlN; TiAlCN; tool; analysis; selected area diffraction (SAED); electron energy loss spectroscopy (EELS)





## 1. Introduction

Surface engineering includes all surface modifications of the material, from ordinary painting, grinding, etc., to the smallest interventions in the atomic structure of the surface, including the application of thin layers on a surface with a thickness of one atom [1]. Methods to modify the nanostructure of the surface of material include, for example, the application of thin films using plasma, ion bombardment, self-assembly, nano-processing, chemical processing, and other processes [2–5]. Surface engineering techniques are currently used in virtually all sectors of industry, including the automotive and aerospace industries, the gun industry, the energy industry, the electrical industry, biomedicine, the textile industry, the oil industry, the chemical industry, the steel industry, the machinery industry, and the construction industry. These techniques are used to develop a wide range of advanced functional properties such as physical, chemical, electrical, magnetic, mechanical properties, wear resistance, and anti-corrosion properties for the surfaces of the desired products [6–10]. Almost all materials, such as metals, ceramics, polymers, and mixtures, can be deposited on the same or other materials [11]. Advanced methods today include CVD (chemical vapor deposition) and PVD (physical vapor deposition). The CVD method uses for the deposition of a mixture of chemically reactive gases ($TiCl_3$, $CH_4$, $AlCl_3$, etc.) heated to a relatively high temperature of 950–1050 °C [12]. PVD technology is based on physical principles-vacuum evaporation or vacuum sputtering of materials contained in the coating (Ti, Al, Si, Cr, etc.), ionization, and their subsequent application to tools at temperatures below 500 °C [12,13].

The high-power impulse magnetron sputtering (HIPIMS) method is a relatively new coating method, but it achieves better properties for coated instruments in some applications than it did for conventional gas phase sputtering methods. The HIPIMS method has

recently emerged as one of the most widely used methods for the deposition of nitride-based coatings. It is a method that allows the formation of thin layers (1.5 μm) [14], which allows a large space for their combination. Due to the low working temperature, below 600 °C, this method does not pose the risk of thermal influence on the material. Originally it was used for coating tools made of high-speed steel. Today it is also used for cemented carbide tools [15–17]. The method has a directional effect, which means that in order to achieve a uniform coating on the entire substrate, it must be rotated during the coating process. There is talk of a so-called shadow effect. Unlike the CVD method, HIPIMS allows coating of even sharp edges, as well as deep-drawing tools: an edge radius of less than 2 μm is sometimes undesirable because of possible breakage of the coating due to insufficient support of the substrate. The radius of the edge strongly influences the cutting process [18]. The advantages of HIPIMS compared to DC magnetron sputtering include, in particular, up to 100 times the electron density, higher ionization of the sputtered particles, and a large number of high-energy ions [19]. Furthermore, coatings applied by the HIPIMS method achieve better wear resistance, a wear coefficient of $4.4 \times 10^{-16}$ $m^3$ $N^{-1}$ $m^{-1}$ for TiAlCN/VCN coating [20]. In addition, the high ionization of the sputtered material makes it possible to control the development of the microstructure and adapt the coating properties as needed [21]. However, before the coating itself, the substrate must undergo a much more thorough cleaning than with classical coating [22,23].

The HIPIMS method, in which the deposited particles are target ions, with ion energy in the ion flux in the range of 10–20 eV [24,25], produces high-density layers with a very fine surface in terms of its morphology. Coatings can be applied as a monolithic layer, coatings consisting of multiple layers of different chemical compositions (multilayer) [26,27], with continuously changing composition (gradient layer) [28,29] or, in the case of new technologies, thin layers up to 10 nm (so-called nanoscales) [30,31]. Particles (atoms, or clusters of atoms) released from sources (so-called "particles") are then ionized and react with inert (usually Ar) and reactive gases inside the chamber (O and N) [32,33]. The coatings are formed at medium to high vacuum, which means at a pressure of about 0.1–1 Pa [34]. With this method of deposition, carbides, borides, and nitrides of transition metals form a more coherent interface with steel in comparison with elements that bind to metal materials ionic or covalently. As already mentioned in work by Holleck [35], transition metals such as Mo, Cr, and W have increased solubility in iron. During the experiment, it was found that these elements can form either a well-mixed and/or graded interface, providing higher adhesion to the steel surface. Chowdhury [36] mentioned in his work that the improvement of adhesion of the C-based coating on steel substrates can be achieved by carbonization or nitriding of the steel surface (substrate). During the experiment, it was found that carbonization saturates the surface of the steel to act as a diffusion barrier. Thus, the result is an improvement in surface roughness, which improves adhesion. Similar results were achieved in his work by Ehiasarian et al. [37]. During the experiment, the steel substrate was bombarded with Cr and Nb ions at a temperature of 450 °C, which caused increased surface mobility of incoming ions on the surface of the substrate and promoted the deposition of metal (crystalline) interlayers and the formation of coherent interfaces. As a result of the bombardment of the surface, the contaminant was removed and, consequently, the formation of mixing zones and the growth of crystalline metal interlayers, which greatly increased the adhesion of the coating to the substrate. Another result was that as a result of these microstructural changes, adhesion at the interface between the metal interlayers and the steel substrate increased.

HIPIMS technology is also suitable for coatings such as TiAlN or TiAlCN, which are characterized by high hardness, good thermal stability, and high corrosion resistance [38–40]. These coatings combine the mechanical properties of TiAlN and TiCN together with the presence of free carbon depending on the amount of carbon in the structure of TiAlN [40,41]. Several authors, for example, used the so-called mosaic TiAl-graphite targets for the deposition of superhard TiAlCN coatings by radio frequency magnetron sputtering (40 GPa) to obtain a nanocomposite structure consisting of FCC-TiAlCN, HCP-AlN crystals and

amorphous carbon [42]. In addition, tribomechanical studies at temperatures up to 600 °C have also been performed in TiAlCN coatings coated with inductively coupled plasma (ICP) on high-speed steel, which showed that high-carbon TiAlCN retained its hardness even after high-temperature annealing [43]. Recently, two important works have been published by Sahul et al. [44] and Rashidi et al. [45] concerning the deposition and analysis of TiAlCN structures. Sahul et al. [44], in their work, used cathodic arc evaporation and $C_2H_2$ as a precursor gas to apply TiAlCN coatings to cemented carbide substrates. As a result, a significantly lower coefficient of friction was obtained for coatings containing 18.7 and 22.3 at.%. On the other hand, they used lateral rotating cathode arc (LARC) technology for the deposition of TiAlCN coatings. As with previous researchers, the resulting TiAlCN coatings consisted of FCC-TiAlN, HCP-AlN, and amorphous carbon phases, which acquired high wear resistance and reached a friction coefficient of 0.17.

The HIPIMS method is a progressive coating method that allows you to combine different coating materials while providing optimal flexibility and hardness of the coating. Therefore, extensive research and analysis of the newly developed layers of TiAlCN and TiAlN were carried out. These layers were invented at the Faculty of Mechanical Engineering at UJEP in cooperation with the company Měd Povrly (Povrly, Czech Republic), manufacturer of semi-finished products of copper and brass) and have not yet been studied in a similar study. The aim of this work was to explore and analyze the composite layers of TiAlN and TiAlCN of the new composition. Eliminating defects, improving the quality of the material, and examining the interconnectedness of the layers can consist in examining all the factors affecting this process. In particular, it is about mapping the thickness of the layers and their connectivity with the underlying substrate. An increase in the surface wear resistance and an increase in the abrasion resistance of the resulting product can be achieved by examining factors affecting the HIPIMS process, such as the connectivity and compactness of the layers on the surface of the base material. The possibilities of using these thin layers are wide, such as anti-reflective coating of optical elements, sputtering of mirror layers, or coating of cutting tools.

## 2. Material and Methods

As an experimental material, steel STN 14109 was used, refined to 56 + 2HRC, its chemical composition is: C-0.98%, Mn-0.3%, Si-0.2%, P-0.027%, Cr-1.45%. This material has good heat ductility, high machinability, high corrosion resistance, and tensile strength higher than 680 MPa. Some of the properties and permissible chemical compositions of the used material are shown in Table 1.

**Table 1.** Permissible chemical composition and mechanical properties of used steel, specified as STN 14 109 standard [46].

| C, wt.% | Mn. wt.% | Si, wt.% | Cr, wt.% | P, wt.% | S, wt.% | Cu + Ni, wt.% | Fe |
|---|---|---|---|---|---|---|---|
| 0.9–1.2 | 0.3–0.5 | 0.15–0.35 | 1.3–1.65 | Max. 0.027 | Max. 0.03 | Max. 0.5 | Reduce |
| $Rp_{0.2}$, MPa | | | | 441—mean value | | | |
| $Rm$, MPa | | | | 608–726 | | | |
| $A_5$, % | | | | 18 | | | |
| Z, % | | | | 35 | | | |

$Rp_{0.2}$—yield strength; $Rm$—tensile strength; $A_5$—elongation; Z—contraction.

The samples were prepared in the form of deep drawing tools intended for production of cartridges (Figure 1).

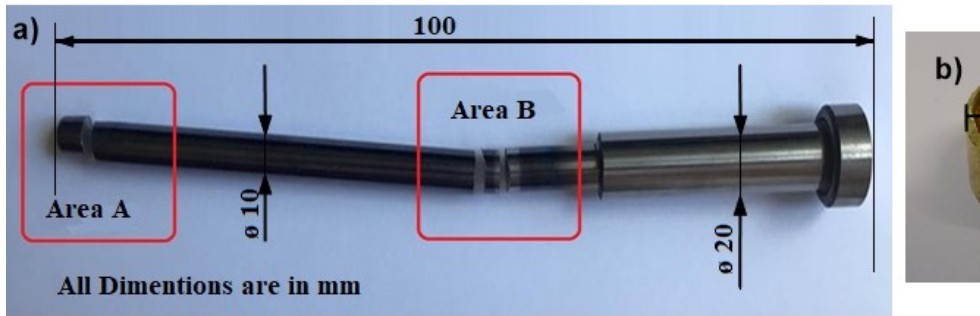
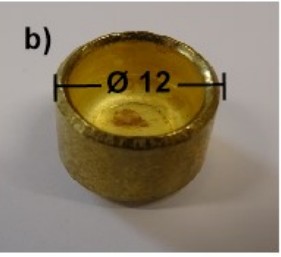

**Figure 1.** Deep drawing tool (**a**) for production of cartridges (**b**).

## 3. Experimental Setup

A two-layer coating was made on the instrument by the PVD method using magnetron sputtering (HIPIMS). The working parameters are shown in Table 2 [47].

**Table 2.** Working parameters for HIPIMS technology.

| HIPIMS Parameter | Value |
|---|---|
| Working pressure | $10^{-4}$–$10^{-2}$ Torr |
| Cathode Current Density | $Jmax \leq 10$ A/cm$^2$ |
| Discharge Voltage | 0.5–1.5 kV |
| Plasma Density | $\leq 10^{13}$ cm$^{-3}$ |
| Cathode Power Density | 1–3 kW/cm$^2$ |
| Ionization Fraction | 30–90% |

The test tool consists of a working zone, which subsequently wears out (Figure 1, area A), and a control zone (Figure 1, area B). Both areas have a lower layer of coating which is TiAlN and top-TiAlCN. In our case, we will consider the areas designated as area a-working zone. The formation of a composite coating with a smooth transition of two layers of different compositions required specific conditions for the formation of the coating so that it was technologically ideal for performing a double layer and, at the same time, achieving the highest possible quality of the coating. The values of the coating formation conditions are in Table 3, with the first column of the table showing the conditions of deposition of the TiAlN coating on the surface of conventional cutting tools. Other columns of the table show the values of applying a composite two-layer deep drawing tool.

**Table 3.** Comparison of coating conditions.

| Thermal Treatment | TiAlN Drill | TiAlCN Composite Bilayer | TiAlN Composite Bilayer | Unit |
|---|---|---|---|---|
| Heating time | 60 | 60 | 140 | min |
| Maximum heating temperature | 708 | 700 | 690 | °C |
| Time to warm up to the highest temperature | 51 | 50 | 45 | min |
| Krypton etching | 60 | 60 | 60 | min |
| Coating temperature | 580 | 580 | 645 | °C |
| Coating time | 110 | 200 | 200 | min |
| Deposition (coating thickness) | $3 \pm 1$ | $3 \pm 1$ | $4 \pm 1$ | µm |
| Cooling | 30 | 30 | 30 | min |

## 4. Experimental Methods

Given the objectives of this part of the research, related to the applied technology of using the tool, two areas were chosen on the tested deep drawing tool: Area A and B (Figure 1). One radial and one transverse sample were taken from each area and prepared. The position of the individual areas was chosen purposefully since different wear values were assumed.

All the samples were prepared manuals by the conventional techniques—wet grinding and after diamond emulsion polishing. The final step of the preparation was polishing with a Struers OP-S suspension. After phosphoric acid etching, the structure of the material was observed and documented by Vega 3 scanning electron microscope (SEM) from Tescan company (Tescan, Vega 3, Brno, Czech Republic). Chemical composition and atomic representation were observed by energy-dispersive X-ray spectroscopy (EDS) X-max analyzer. SDD detector 20 mm², Oxford Instruments, Abingdon-on-Thames, UK. The acceleration voltage of the SEM-EDS method was set to 5 kV.

Aided by a focused ion beam (FIB) on a Jeol microscope from Fei (Hillsboro, OR, USA), the lamellae were prepared for the study of the material composition by EDS in the transmission electron microscope (TEM) JEOL 2200 FS, which is equipped with a camera from TVIPS and EDS from Oxford Instruments. The line profile was measured in a STEM mode with a beam size (spot size) of 1 nm.

## 5. Results and Discussion

### 5.1. Analysis of the Chemical Composition of the Deep Drawing Tool

For the experimental part, steel 14109 was used. According to the standard [48], it is bearing steel with a carbon content of between 0.9% and 1.1%. The highest percentage of content from alloying elements is occupied by chromium, which is represented in steel from 0.5% to 1.5%, and therefore the steel is distinguished by good wear resistance and provides the formation of carbides that increase hardness.

Another very important element is manganese, which is contained in steel from 0.2% to 0.4%, which significantly increases the steel's patency. The chemical composition of the tested steel, measured using the Tasman Q4 spectrometer, is shown in Table 4.

**Table 4.** Chemical composition of steel 14,109 in percentage by weight.

| C | Si | Mn | P | S | Cr | Ni | Mo | Al | Cu | Co | Mg | Sn | Nb | V | Fe |
|---|----|----|----|----|----|----|----|----|----|----|----|----|----|----|----|
| 0.99 | 0.226 | 0.282 | 0.011 | 0.0033 | 1.48 | 0.03 | 0.0078 | 0.042 | 0.021 | 0.011 | 0.0012 | 0.0055 | - | 0.0036 | 96.9 |

### 5.2. Microscopic Evaluation of the Deep Drawing Tool

One of the development groups of the layers was a TiAlN-based layer, which reached high hardness HV up to 33 GPA and had a high maximum working temperature exceeding 900 °C. In addition, these wells exhibit high abrasive strength and heat resistance (chemical stability) at high temperatures. The types of these layers are monolayers, multilayers, gradients, and differently structured TiAlN layers with high hardness and heat resistance. In combination with the additives Hf, Cr, Nb, Si, etc., a very fine-grained and stable structure is obtained. A different Ti:Al ratio in layers is used for specific applications. For example, the increased Al content (about 60%) brings layers with higher resistance to oxidation. Above the content of 65%, only HCP crystallographic lattice is formed, which leads to a deterioration in heat resistance [49,50]. Mechanical properties are improved under temperature load [51].

The next significant generation of coatings is represented by layers on TiAlCN [52,53]. This coating has all the advantages as a variant of the TiAlN coating, while it has one more significant feature, and this is a low coefficient of friction. In wet or dry conditions, this coating will show high performance. These properties of this coating predetermine for versatile use for tools for machining, forming, drawing, etc. [54].

The microstructure of the base material and the deposited TiAlCN + TiAlN layer was examined using an electron microscope with an EDS analyzer was used for a more detailed study of the microstructure. In the first series, an analysis of the unused area A was carried out at a magnification of 2800× (Figure 2). In terms of magnification, the components are not visible, but it is clear that the layer is not flat, without cracks and damage. It can also be stated that the layer is uniform in terms of thickness. On the surface of the layer can be seen black holes that have formed as a result of sample preparation: cutting and grinding. Some

defects were partially smoothed by the polishing process, but not all. No visible difference between the layers was observed. In the investigated area, the thickness of the applied layer was measured, which was continuous and compact and ranged from 5.17 to 5.82 µm. The 2/3 coating consists of a TiAlCN layer corresponding to 3.86 µm and a 1/3 TiAlN layer corresponding to 1.9 µm. In comparison with Chen's study [55], where the value of one of the layers was 3.2 µm, in our case, much thinner layers were achieved, namely about 2.6 µm.

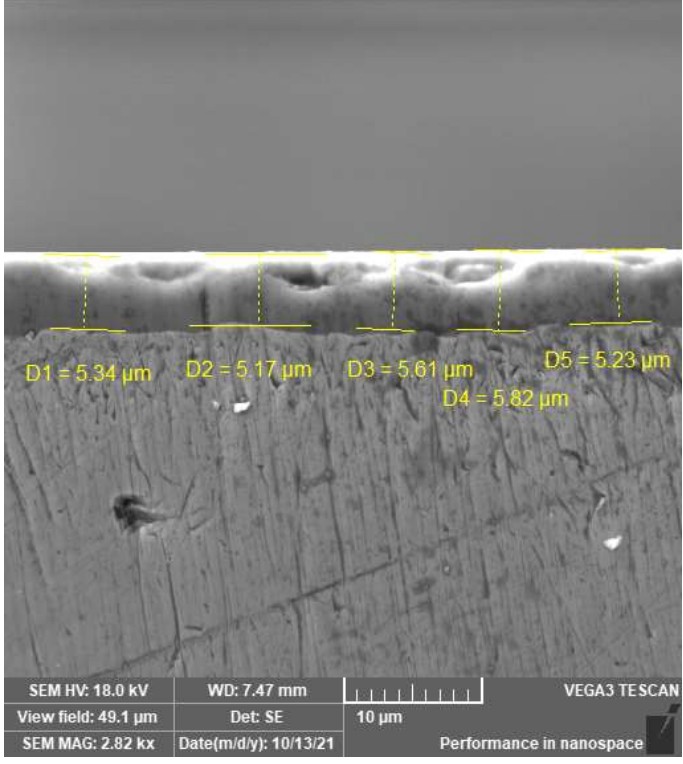

**Figure 2.** A cross section of a micro-coated sample with the measurement of its thickness.

The results of the surface EDS analysis of the selected area (Figure 3b) showed a composite TiAlCN coating with the following percentage of elements: C-39.0%, N-33.9%, Ti-15.9%, Al-10.9%, and a small amount of the Fe-0.3%. The analysis showed that the composite TiAlCN coating 2/3 contains carbon and nitrogen, and 1/3 contains titanium and aluminum. Another investigated region (Figure 3c) showed the presence of an area partly consisting of the TiAlN layer and partly of the substrate with the composition: Ti-28.3%, Al-27.0%, N-23.7%, Fe-12.7%, and C-8.4%. The EDS analysis of the basic material (Figure 3d) was performed with the composition: Fe-87.7%, C-6.6%, Al-2.3%, Ti-1.8%, and Cr-1.5%. As can be seen in Figure 3, the TiAlCN and TiAlN layers were well connected to the base material.

EDS analysis was also performed using a scanning electron microscope, which showed a very uniform distribution of individual elements in a provided layer. EDS elemental mapping was also performed, which showed the distribution of the elements used for coating (titanium, aluminum, and nitrogen), as well as carbon as an alloying element. From the EDS spectra provided in Figure 4, it can be concluded that the distribution of all elements is very equally in the studied area.

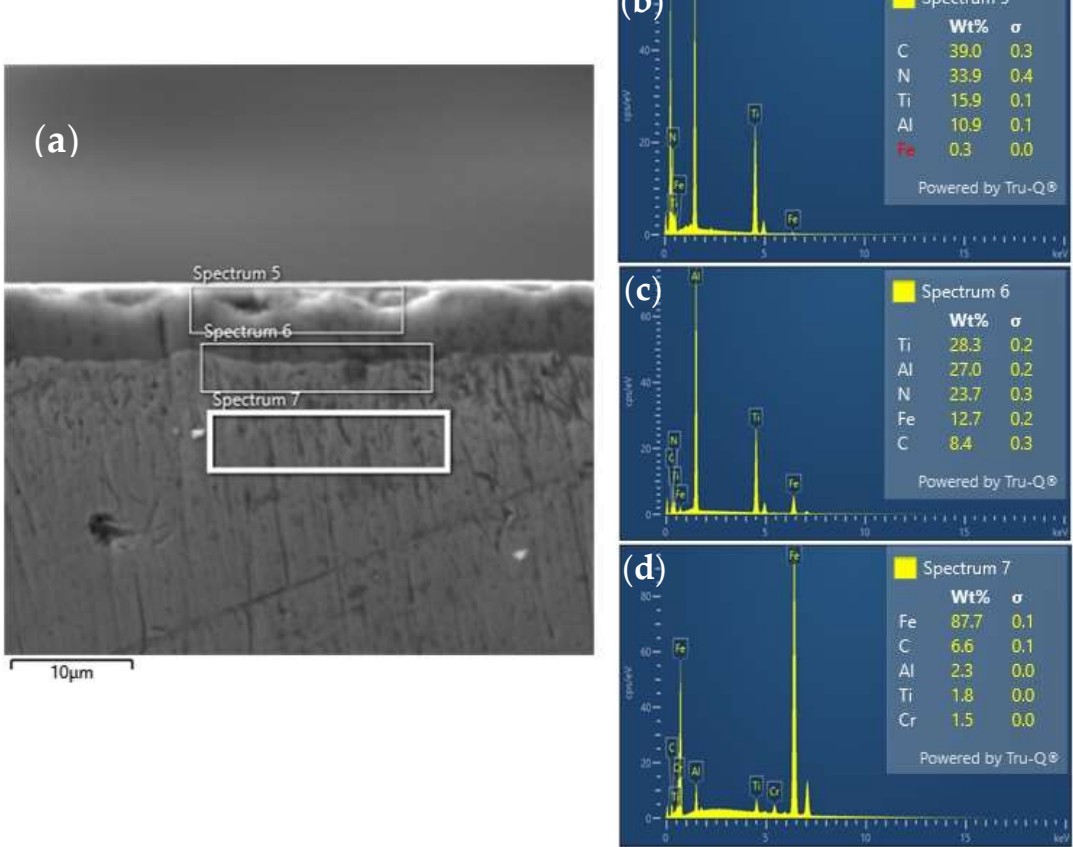

**Figure 3.** Selected site (**a**) for area analysis of the coating with the EDS recording of individual elements: (**b**) EDS spectra of TiAlCN coatings, (**c**) EDS spectra of TiAlCN layer and partly of the substrate, (**d**) EDS spectra of substrate.

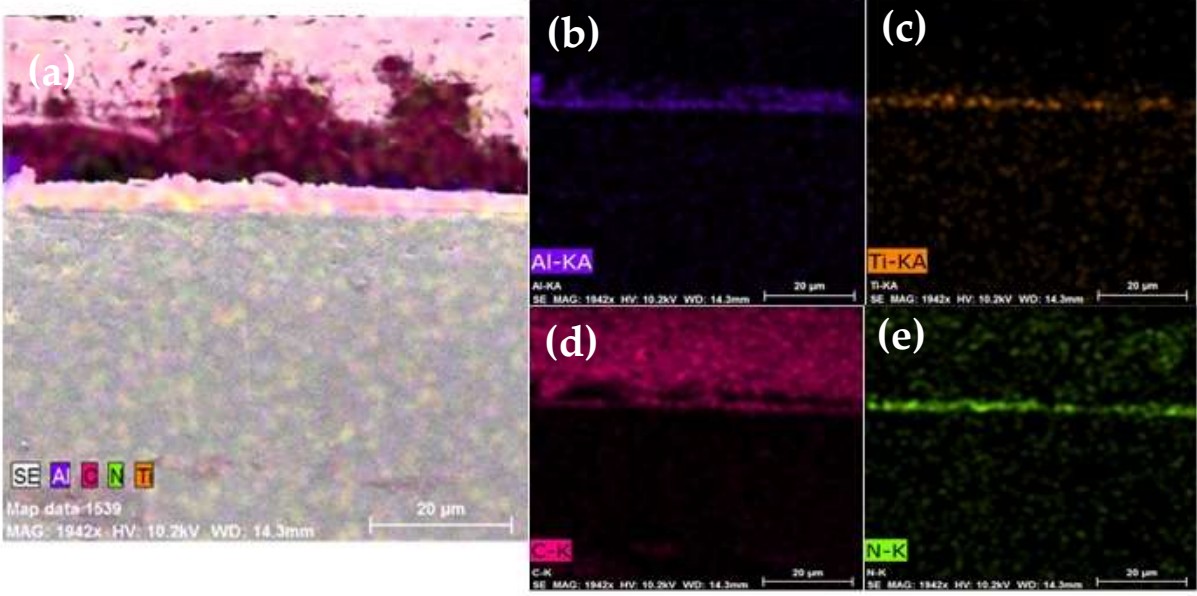

**Figure 4.** EDS analysis of the layout of individual elements in the TiAlCN layer (**a**), elemental distribution of Al (**b**), Ti (**c**), C (**d**) and N (**e**).

### 5.3. Results of TEM Analysis

The FIB-prepared lamella was observed using eftem JEOL 2200 FS (Figure 5). As shown in Figure 6, the lamella was composed of a steel alloy (area A), a coating layer, and platinum covering material. The surface layer is formed by two distinctly separated areas (Figure 6, areas B and C). The area B (sublayer) adjacent to the substrate is formed by grains elongated across the layer. In the second sublayer (area C), the grains were more tender.

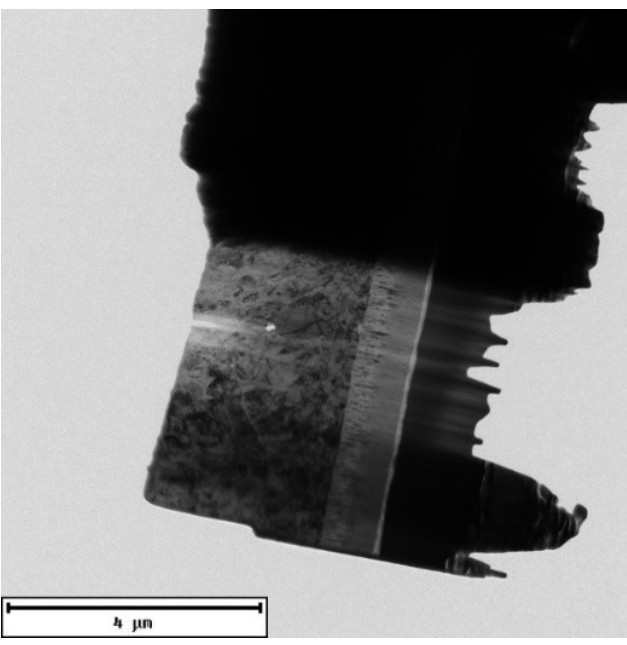

**Figure 5.** Overview of the lamella.

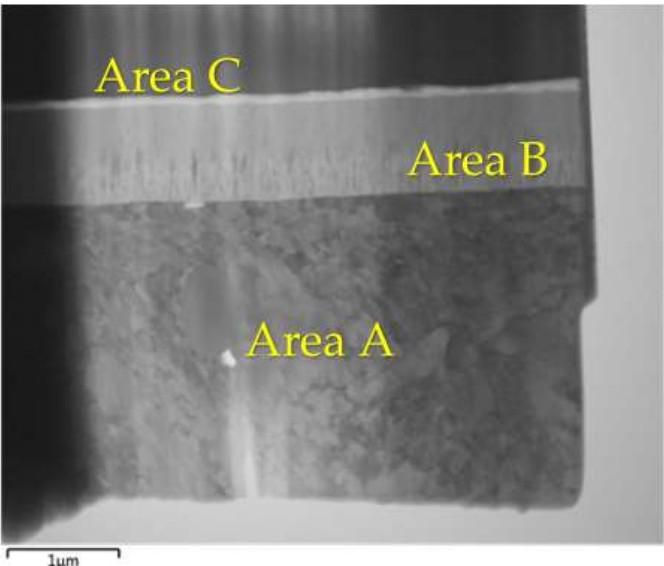

**Figure 6.** Detail of the top and bottom layers of deposited coating.

Perpendicular to the layer, a linear profile was measured using EDS. The analysis was performed in STEM (scanning) mode, and the beam cross-section (spot size) was 1 nm [20]. Figure 7 shows the linear profiles of all detected elements. The concentration of Fe is high in steel and drops very sharply towards the surface. In the layer was the content of Fe at the limit of quantification. Behind the layer, an increase in Fe content is observed, which may be due to contamination. On the outer surface of the layer, an increased concentration

of oxygen was detected, which indicates the presence of a passivation layer (the thickness of the passivation layer is usually about 100 nm, which corresponds here). Surface oxides are mixed Ti and Al. Pt is an artifact from sample preparation. Its presence is noted only behind the oxide layer. This confirms the presence of an oxide layer on the original sample. Ga is used to "carve" the lamella with FIB. Its presence is detected only in the Pt layer. This proves the quality preparation of the lamella, which is not contaminated with Ga.

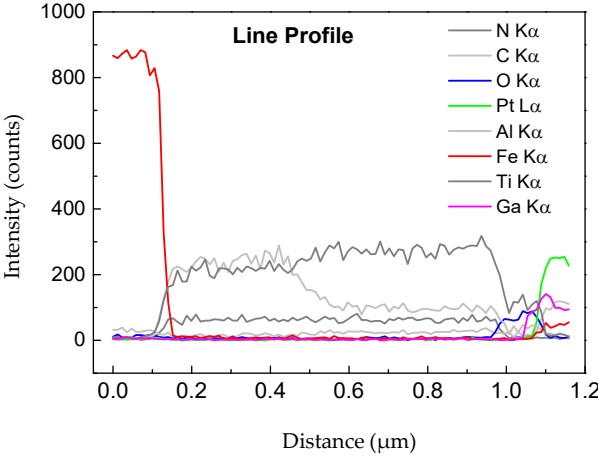

**Figure 7.** EDS linear profile of all detected elements.

In addition to morphology, the composition of the sublayers also differed; in both cases, the sublayers were formed by the intermetallic phase, as shown by the typical "steps" on the EDS profile in Figure 8. The concentration of N was higher in the layer but did not differ for individual sublayers. The concentration of C appears to be higher in the surface sublayer. However, EDS analysis is not very reliable for N and C.

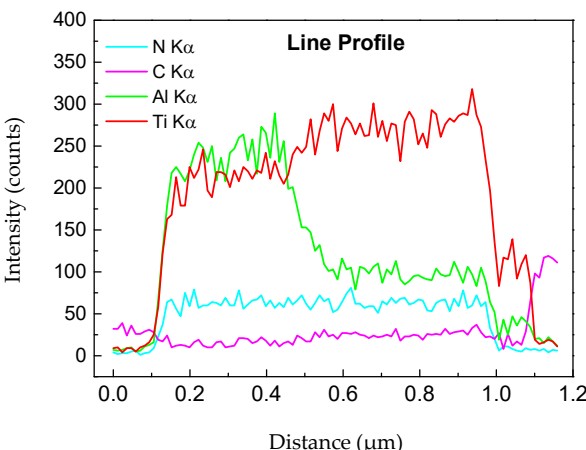

**Figure 8.** EDS linear profile of selected detected elements.

The chemical composition of the sublayers was measured using point EDS analysis, both in the lighter and darker regions of the sublayer adjacent to the substrate and in the surface sublayer (Figure 9). The results in the light and dark areas of the sublayer adjacent to the alloy were identical within the experimental error, and the observed contrast was caused only by diffraction contrast [56,57]. In Table 5, the ratios of Ti and Al are presented.

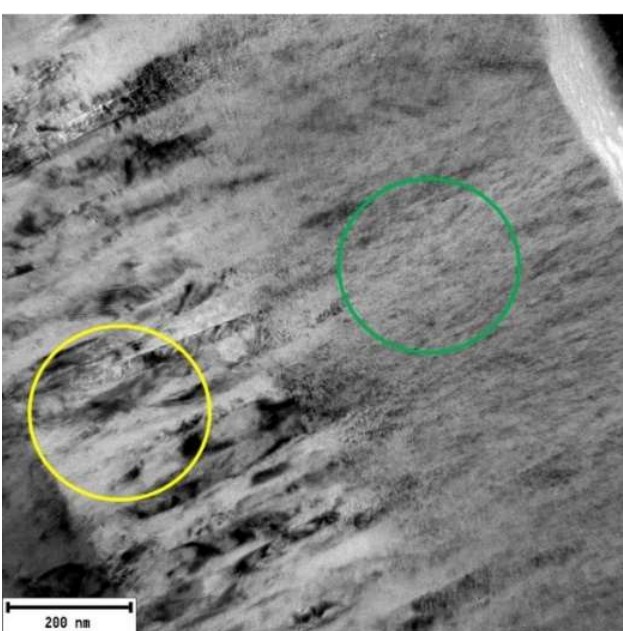

**Figure 9.** Microstructure of the surface layer with marked points of the performed analyses.

**Table 5.** Element content measured by EDS analysis, results reported in at.%.

| Localization of Analysis | Ti | Al | Phase |
|---|---|---|---|
| Sublayer adjacent to the substrate | 55.2 | 44.8 | TiAl |
| Surface sublayer | 71.2 | 28.8 | $Ti_3Al$ |

Further, the structure of both sublayers in the places is indicated in Figure 9. By using electron diffraction, the phase composition was confirmed, which corresponded to the ratios of Ti, and Al indicated in Table 4. The layer adjacent to the substrate is formed by a relatively coarse-grained TiAl phase (marked with a yellow circle). The surface sublayer is formed by the fine-grained $Ti_3Al$ phase (marked with a green circle), as shown in Figures 10 and 11. Diffractograms were evaluated using the crystbox software.

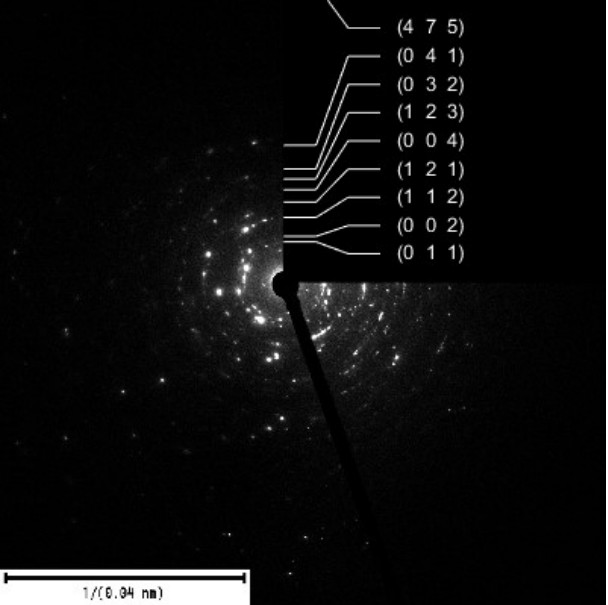

**Figure 10.** Selected area diffraction (SAED) sublayers adjacent to the substrate, indexed by the TiAl phase.

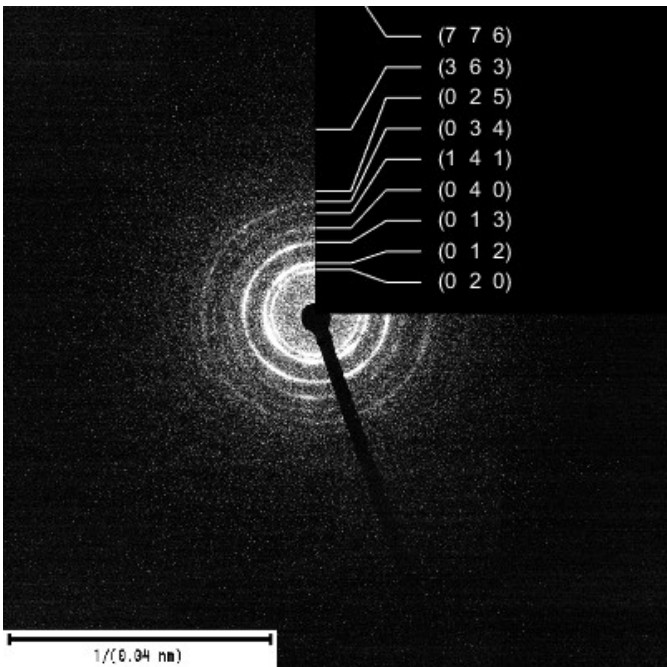

**Figure 11.** Selected area diffraction (SAED) surface sublayers, phase indexed Ti₃Al.

The layer was further examined using electron energy loss spectroscopy (EELS). The Ti-$L_{2,3}$, N-K, and C-K spectra for TiAlN and TiAlCN coatings exhibited peak shapes and positions similar to those of TiC and TiN [58,59]. The presence of C was detected in a very small amount in the sublayer adjacent to the alloy, see Figure 12.

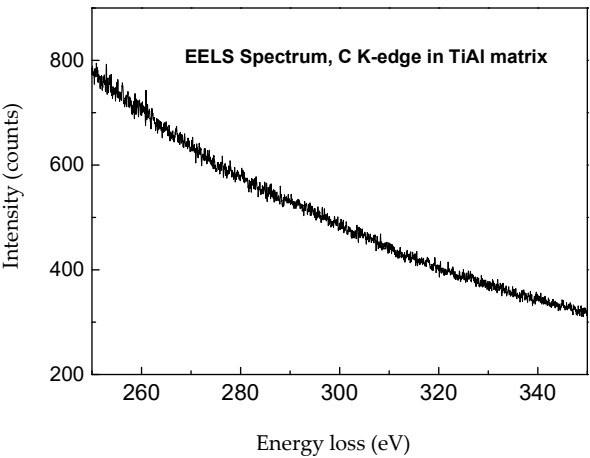

**Figure 12.** EELS spectrum in the sublayer adjacent to the substrate, K Edge C (284 eV).

On the contrary, the presence of a very small amount of C was detected in the surface sublayer, as shown in Figure 13. These results are consistent with the EDS profile in Figure 7. The shape of the Ti peak is comparable in both cases (Figures 14 and 15). On the contrary, the shape of the peak N differs significantly. The primary explanation may be in different binding conditions N (one peak at TiAl would indicate a more metallic bond, and a bifurcated peak at Ti₃Al would correspond to a more covalent bond). In Ref. [18], Lengauer et al. have observed a similar phenomenon. The ratio of Ti and N elements can be evaluated from the ratio of the areas of the individual peaks, and the results are shown in Table 6. For comparison, the ratios of the contents from the EDS analysis are also provided. The results are in surprisingly good agreement, although the quantification of N from the EDS analysis may not be accurate.

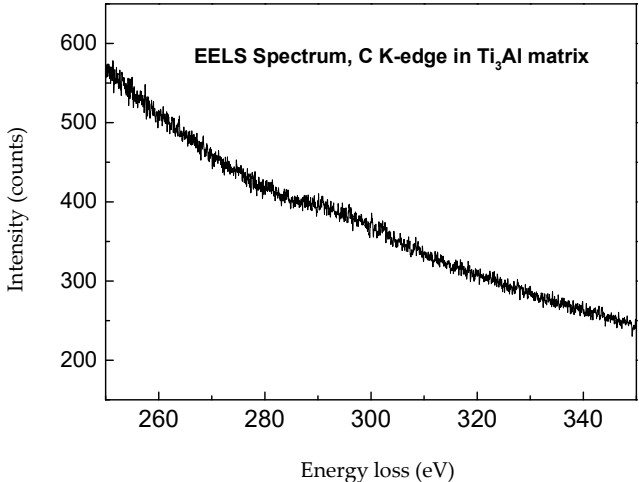

**Figure 13.** EELS spectrum in surface sublayer, K Edge C (284 eV).

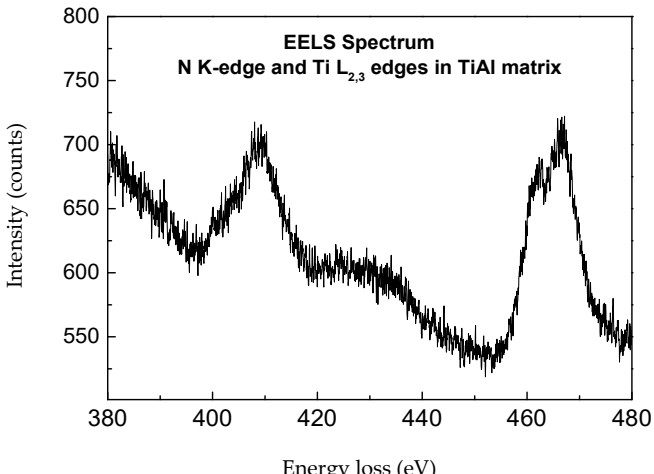

**Figure 14.** EELS spectrum in the sublayer adjacent to the substrate, K Edge N (399 eV) and L edge Ti (455 eV).

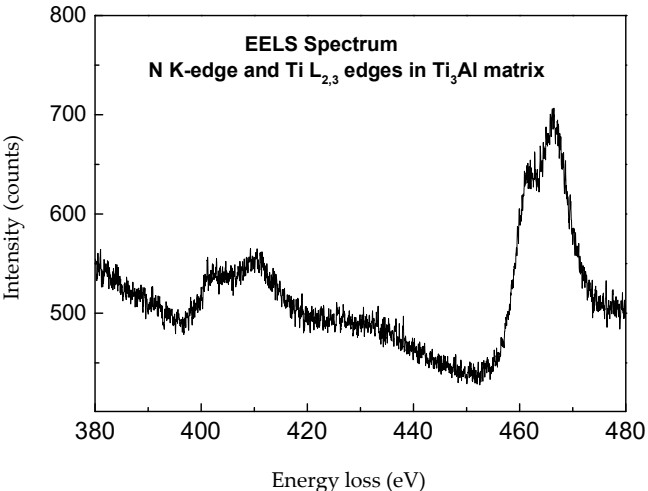

**Figure 15.** EELS spectrum in surface sublayer, K Edge N (399 eV) and L edge Ti (455 eV).

**Table 6.** Ti/N element content ratio, results reported in at.%.

| Localization of Analysis | EELS | EDS |
|---|---|---|
| Sublayer adjacent to the substrate | 1.73 | 1.73 |
| Surface sublayer | 2.40 | 2.66 |

*5.4. Analysis of the Deep Drawing Tool after Wear*

After the deep drawing tools were coated, the coating was tested in operation. The deep drawing tool was put into operation in the metalworks of the company Měd Povrly, and it was subjected to production strokes for the production of the cup. The mandrel, which was used in the production sphere until now, reached the number of 365,000 strokes, and it was necessary to replace it. After the subsequent addition of the deposited composite two-layer TiAlCN + TiAlN reached about twice the service life, namely—702 381 production strokes. After the tool had been worn, a layered TiAlCN + TiAlN analysis was subsequently performed. Figure 16 (right) shows the place with the expected maximum wear. This is the place of contact at the edge of the cartridge case. In Figure 16 on the left, you can see the delaminated deposited layer just at the point of greatest wear.

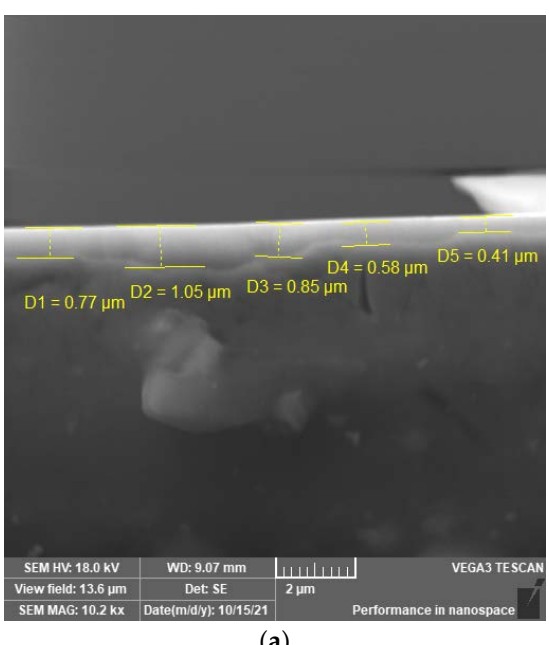

(**a**)

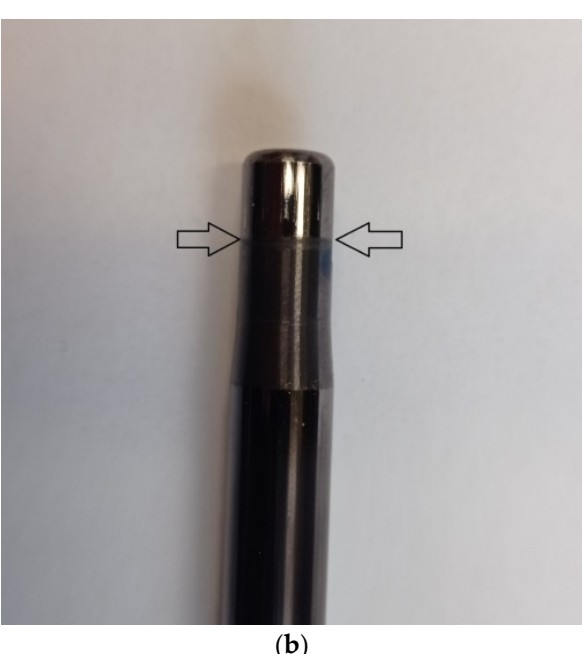

(**b**)

**Figure 16.** SEM image of the delaminated layer (**a**) and place of deepest tool damage (**b**).

When assessing the thickness of the layer using electron microscopy, it was found that there was a loss of the layer up to 0.85 μm. Microscopic analysis of the worn layer using SEM microscopy indicates a thinning of the deposited layer. This thinning is due to the high number of working cycles. The subsequent EDS analysis (Figure 17) shows a worn top layer of TiAlCN with the following chemical composition C-68.7%, N-17.5%, Al-9.7%, and Ti-3.0%. When comparing the unused and worn tools, the Ti content was reduced by more than 14%. The analyses show that the coating applied to the tool (cartridge) shows much better performance in the production of the cup (up to 200%). The main advantage of this coating is that even after the subsequent machining of the tool, it is possible, even after so many cycles, to coat the tool again and immediately put it into circulation.

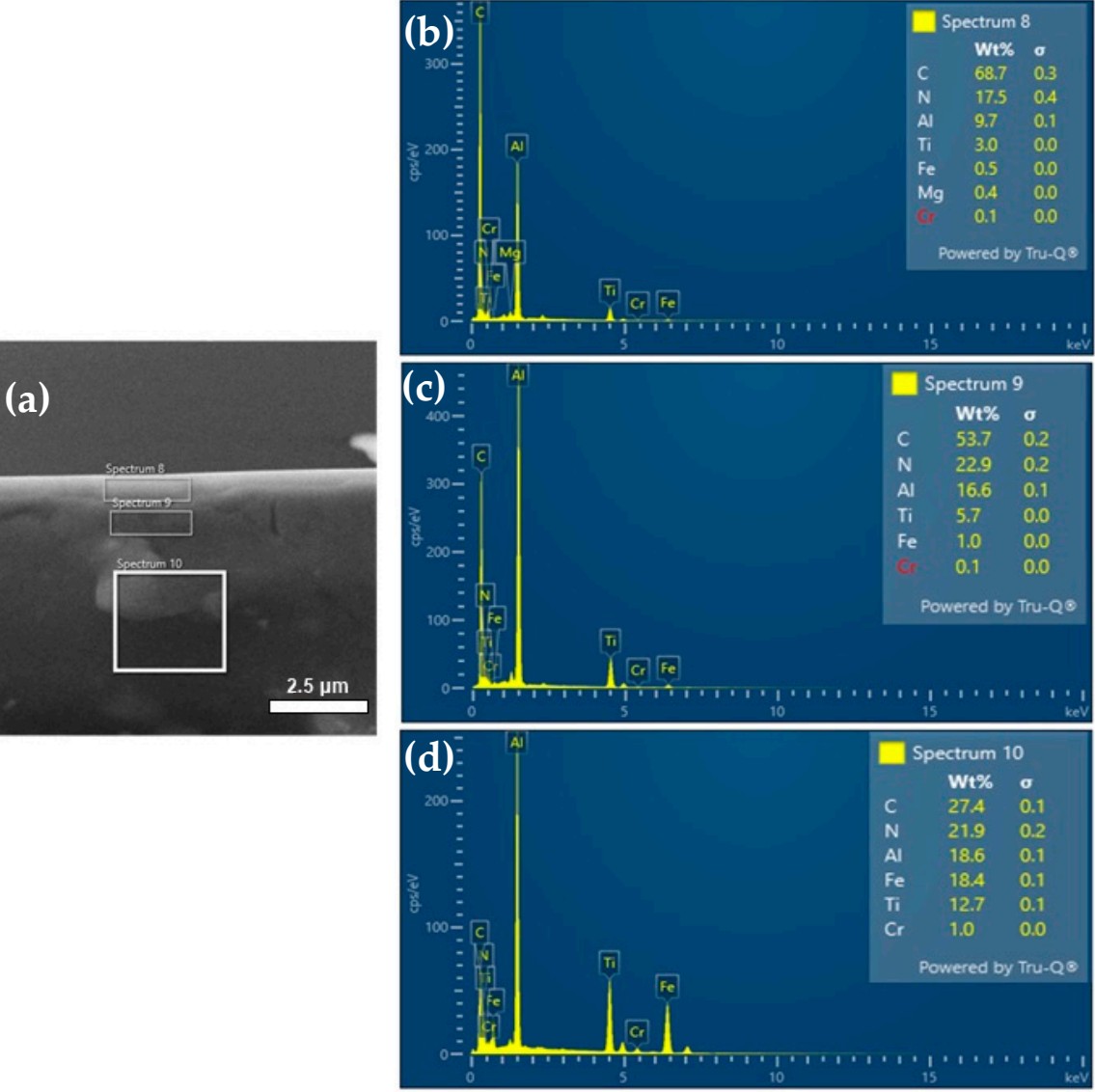

**Figure 17.** Selected site (**a**) for area analysis of the coating with the EDS recordings of individual elements: (**b**) EDS spectra of TiAlCN coatings, (**c**) EDS spectra of TiAlCN layer and partly of the substrate, (**d**) EDS spectra of substrate.

## 6. Conclusions

The research carried out in this publication was aimed at the development and research of a new multilayer micro-coating for a deep-drawing tool designed for drawing cartridges, and the following conclusions were reached:

- The spectral analysis performed showed that the basic material of the deep-drawing tool is STN 14 109 steel;
- The analysis of surface morphology showed a good and high-quality diffusion connection between the base layer and the coating, where the coating is continuous, compact, and without defects, it increased porosity, microcracks, etc. The applied layer ranges from 5.2 to 5.8 µm;
- The subsequent EDS analysis showed the homogenization of the individual elements and their uniform representation in a provided layer: 2/3 contains carbon and nitrogen, and 1/3 contains titanium and aluminum. In addition, selected area diffraction (SAED) demonstrates that the presence of carbon was not detected in the sublayer adjacent

to the base material. In contrast, the presence of a very small amount of carbon was detected in the surface layer of the coating;

- TEM-produced lamellas showed the presence of two layers, namely: the first area (sublayer) adjacent to the substrate is formed by grains elongated across the layer; in the second sublayer, the grains were finer,
- Electron diffraction confirmed the phase composition, where the layer adjacent to the substrate is formed by the relatively coarse-grained phase TiAl, and the surface sublayer is formed by the fine-grained phase $Ti_3Al$;
- ELLS analysis showed that a single peak at TiAl would indicate a more metallic bond, and a bifurcated peak at $Ti_3Al$ would correspond to a more covalent bond;
- The EDS analysis of the worn tool showed, in addition to the thinning of the deposited layer to a value of 0.85 μm, also a decrease in the Ti content by more than 14%.

From the above conclusions, it can be judged that HIPIMS technology will guarantee the required excellent homogeneity in the distribution of individual elements in the coating, as well as uniform thickness of the coating. In addition, the use of a new multilayer micro-coating significantly increased labor productivity and cup production by 200%.

**Author Contributions:** Conceptualization, J.N. and Š.M.; methodology, I.H.; software, I.H.; validation, J.N., Š.M. and I.H.; formal analysis, S.L.; investigation, J.N.; resources, I.H.; data curation, I.H.; writing—original draft preparation, J.N.; writing—review and editing, S.L.; visualization, I.H.; supervision, Š.M.; project administration, Š.M.; funding acquisition, Š.M. All authors have read and agreed to the published version of the manuscript.

**Funding:** Supported by the OP VVV Project Development of new nano and micro coatings on the surface of selected metallic materials-NANOTECH ITI II, Reg. No CZ. 02.1.01/0.0/0.0/18_069/0010045.

**Institutional Review Board Statement:** Not applicable.

**Informed Consent Statement:** Not applicable.

**Data Availability Statement:** Data sharing is not applicable to this article.

**Conflicts of Interest:** The authors declare no conflict of interest.

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
