# Peer review of "Analysis of Composite Coating of Deep Drawing Tool"

_coatings, doi:10.3390/coatings12060863_

Round 1

Reviewer 1 Report

The paper entitled "Analysis of composite coating of deep drawing tool" presented R&D works on multi-layer micro-coating for a deep-drawing tool. Have gone through the manuscript and found the authors presented relevant experimental data and analysis. The paper is in good shape (Some English changes are required) and a few minor comments are below to improve the paper.
Abstract: Remove a couple of lines (should go to the introduction) and write what was done and what was achieved.
The introduction (First paragraph) needs a couple of references to support all lines.
EELS spectrums: Please provide some information on the peaks, shape, etc. with some references.
Line 265-267: "The results in the light and dark areas of the sublayer adjacent 265
to the alloy were identical within the experimental error, and the observed contrast was 266
caused only by diffraction contrast". Please provide a reference.

Author Response

Thank you so much for your comments and your time, I have removed a few lines from the abstract to introduction and have changed it. Also I have add a new references to the introduction to support all lines. Add the information of the peaks, shape of EELS spectrum and provide a reference to the rest o fit.

Reviewer 2 Report

Reviewer’s Report on Ms with ID: coatings-1743757

Title: Analysis of composite coating of deep drawing tool

by Jan Novotný, Iryna Hren, Štefan Michna, Stanislaw Legutko

The research work presented in this article is very relevant and the results are of interest for many branches in the industry. The authors have developed composite layers of TiAlN and TiAlCN with new composition, deposited on steel STN 14109 substrates by high power impulse magnetron sputtering (HIPIM) technique. By applying modern and powerful microscopic methods, the authors have conducted extensive research and analysis on the as-deposited and tested in operation samples. The experimental results are presented qualitatively in a comprehensive way. However, in the manuscript I found mistakes that need to be corrected before it is accepted for publication.

 My remarks and recommendations are as follows:

 - It is accepted that the abbreviation is placed after the name of the subject, i.e. “The high power impulse magnetron sputtering (HIPIMS) method.....”

- Page 2, line 52; “...the formation of very thin layers (1.5 μm) [4,5],...” ; a 1.5 μm thick layer is not “very thin”.

- It may be more correct to write: “a Struers OP-S suspension” instead of “a struers ops supension” (Page 4, line 163) and, “camera from TVIPS” instead of “camera from tvips” (Page 5, line 171).

- Page 5, lines 202-204; The sentence “The microstructure of the base material and the deposited TiAlCN + TiAlN layer was examined using a tescan VEGA 3 XMU electron microscope with an Oxford EDS analyzer was used for a more detailed study of the microstructure” needs revision. Besides, the type of microscope used is given in section 4, where it should be. Here, it is a needless repetition.

- Page 7, lines 223-228; The sentences related to EDS results shown in Fig. 4 should be revised. A possible suggestion is: EDS elemental mapping was also performed which showed the distribution of the elements used for the coating (titanium, aluminum and nitrogen), as well as carbon as an alloying element. From the EDS spectra, given in Fig. 4, it can be concluded that the distribution of all elements is very homogeneous in the studied area.

- Page 9; The denotation of AREA C is missing in Fig. 6. 

- In Figs. 7,8,10,11 the titles and dimensions of the axes should be as follows: Intensity (counts), Distance (mm) and Energy loss (eV) instead of Intensity/Counts, Distance/mm and Energy loss/eV.

- The number of Fig. 8 is given twice, one showing the linear profile of the EDS of the detected elements, and the second showing a cross-sectional TEM image of the studied layer. In addition, it would be more informative if the plane of the substrate surface is marked on the TEM image.

-   Page10, lines 269-274; In Fig. 8 (the first one) it is not clear which plane is marked with zero, whether it is the substrate surface or the top of the deposited layer. From the related text it can be concluded that the zero indicates the surface of the substrate, but if so, it should be mentioned in the text. Moreover, the figures’ numbering should be corrected in this text.

- The following sentences contradict each other: “The layer was further examined using electron energy loss spectroscopy (EELS). The presence of C was not detected in the sublayer adjacent to the alloy, see Fig. 11.” (lines 282-283) and “On the contrary, the presence of a very small amount of C was detected in the surface sublayer, as shown in Figure11.” (lines 288-289)

- Page 13, Lines 299-300; ”A similar phenomenon was observed in his work Lengauer [44]” needs revision. A suggestion is: “......in the work of Lengauer [44]  or  In ref [44] Lengauer et al. have observed a similar phenomenon

In Ref.8 the year of presentation is missing.

Author Response

Thank you so much for your comments and your time. Those comments are all valuable and very helpful for revising and improving our paper, as well as the important guiding significance to our researches. I have changed the abbreviation, some of the expression, have changed the sentence “The microstructure of the base material“ and “The sentences related to EDS results shown“. Also I have fixed all figures, also its numbering, unify the information about The presence of C in the sublayer. I have change the sentence “In ref [44] Lengauer et al. have observed a similar phenomenon”and add the missing year of the ref. 8.

Reviewer 3 Report

The article reports on thin coating micro-layers TiAlN and TiAlCN, prepared by HiPIMs coating technology. The chemical composition and the microstructure analyses were reported using SEM and TEM characterization. The paper reveals that the multilayer microcoating based on TiAlN+TiAlCN with a thickness of 5.8 µm increases the repeatability of production strokes 20 by 200 %.

The paper describes the preparation of the coatings using HiPIMS technology. It focusses on the study of hard multilayers for increasing the wear performance of the tools. However, the study lacks an in-depth analysis and critical vision. First, no deposition parameters are provided? What is the pressure? Why only HiPIMS? What about the DCMS? What is the thickness of each layer even 5.8 µm gives better result, is there any effect of their thickness on the tool’s performance?

Line 57: What authors means by “HiPIMS allows even sharp edge? What is the configuration of the machine? experimentally how these layers are obtained?

I cannot recommend further consideration for publication if authors do not provide critical study as mentioned above.    

Other comments:

Line 193, please give references.       

The quality of some images is not good, EDS Fig. 3. The scale font should be improved, Fig. 5. Is it TEM or SEM image?

Fig 6. There is no Area C.

Line 244: the sentence is not clear on Fe content?

Line 289: Fig. 12?   

Give the energy values or indicate the peaks on Fig 13 and 14.

Line 299/300: please rephrase.

Line 342: what mean by “homogenization of the individual element”?

Author Response

 Thank you so much for your comments and your time. Those comments are all valuable and very helpful for revising and improving our paper, as well as the important guiding significance to our researches. I have add the working parameters to the main text.

The 2/3 coating consists of a TiAlCN layer corresponding to 3.86 µm and a 1/3 TiAlN layer corresponding to 1.9 µm.

Unlike the CVD method, HIPIMs allows coating even sharp edges, as well as deep-drawing tools: with an edge radius of less than 2 μm, is sometimes undesirable because of possible breakage of the coating due to insufficient support of the substrate. The radius of the edge strongly influences the cutting process [18]. The advantages of HiPIMS compared to DC magnetron sputtering include, in particular, up to 100 times the electron density, higher ionization of the sputtered particles and a large amount of high-energy ions [19]. Furthermore, coatings applied by the HiPIMS method achieve better wear resistance, a wear coefficient of 4.4.10-16 m3N-1m-1 for TiAlCN / VCN coating [20].

I also have add this to the text. Thank you

Also thank you for the rest of the comments. I have add the missing references, improve the quality of the pictures, add the missing information to the Figure 13 and 14. Add the missing information and fixed the sentence of the Fe content, the ratio of Ti and N elements and “homogenization of the individual element”.

Thank you so much for your comments and for your time.

Round 2

Reviewer 3 Report

Authors responded to all comments and improved the manuscript. The manuscript now can be accepted for publication in Coatings. 

Author Response

Thank you so much for your comments and your time. Those comments are all valuable and very helpful for revising and improving our paper, as well as the important guiding significance to our researches. Your are right, there no Fe in the layer. I have fixed it. Also in terms of magnification, the components are not visible, but it is clear that the layer is not flat, without cracks and damage. It can also be stated that the layer is uniform in terms of thickness. On the surface of the layer can be seen black holes that have formed as a result of sample preparation: cutting and grinding. Some defects were partially smoothed by the polishing process, but not all. I have add those comments also to the text.

Thank you so much for your comments and for your time.
